# Pilot Study for Correlation of Heart Rate Variability and Dopamine Transporter Brain Imaging in Patients with Parkinsonian Syndrome

**DOI:** 10.3390/s22135055

**Published:** 2022-07-05

**Authors:** Devdutta S. Warhadpande, Jiayan Huo, William A. Libling, Carol Stuehm, Bijan Najafi, Scott Sherman, Hong Lei, Janet Meiling Roveda, Phillip H. Kuo

**Affiliations:** 1Department of Medical Imaging, Banner University Medical Center, University of Arizona, Tucson, AZ 85724, USA; pkuo@email.arizona.edu; 2Department of Biomedical Engineering, University of Arizona, Tucson, AZ 85721, USA; jiayanh@email.arizona.edu (J.H.); meilingw@arizona.edu (J.M.R.); 3Arizona College of Osteopathic Medicine, Midwestern University, Glendale, AZ 85308, USA; william.libling@midwestern.edu; 4Department of Medical Imaging, University of Arizona, Tucson, AZ 85724, USA; carols@arizona.edu; 5Division of Vascular Surgery and Endovascular Therapy, Department of Surgery, Baylor College of Medicine, Houston, TX 77030, USA; bijan.najafi@bcm.edu; 6Department of Neurology, University of Arizona, Tucson, AZ 85724, USA; ssherman@neurology.arizona.edu (S.S.); hong.lei@bannerhealth.com (H.L.); 7Department of Electrical and Computer Engineering, University of Arizona, Tucson, AZ 85721, USA; 8Department of Medicine, University of Arizona, Tucson, AZ 85724, USA

**Keywords:** HRV, Parkinson disease, heart rate variability, I-123 ioflupane, DaTscan, autonomic dysregulation, autonomic dysfunction, neurodegeneration

## Abstract

Background: Parkinsonian syndrome (PS) is a broad category of neurodegenerative movement disorders that includes Parkinson disease, multiple system atrophy (MSA), progressive supranuclear palsy, and corticobasal degeneration. Parkinson disease (PD) is the second most common neurodegenerative disorder with loss of dopaminergic neurons of the substantia nigra and, thus, dysfunction of the nigrostriatal pathway. In addition to the motor symptoms of bradykinesia, rigidity, tremors, and postural instability, nonmotor symptoms such as autonomic dysregulation (AutD) can also occur. Heart rate variability (HRV) has been used as a measure of AutD and has shown to be prognostic in diseases such as diabetes mellitus and cirrhosis, as well as PD. I-123 ioflupane, a gamma ray-emitting radiopharmaceutical used in single-photon emission computed tomography (SPECT), is used to measure the loss of dopaminergic neurons in PD. Through the combination of SPECT and HRV, we tested the hypothesis that asymmetrically worse left-sided neuronal loss would cause greater AutD. Methods: 51 patients were enrolled on the day of their standard of care I-123 ioflupane scan for the work-up of possible Parkinsonian syndrome. Demographic information, medical and medication history, and ECG data were collected. HRV metrics were extracted from the ECG data. I-123 ioflupane scans were interpreted by a board-certified nuclear radiologist and quantified by automated software to generate striatal binding ratios (SBRs). Statistical analyses were performed to find correlations between the HRV and SPECT parameters. Results: 32 patients were excluded from the final analysis because of normal scans, prior strokes, cardiac disorders and procedures, or cancer. Abnormal I-123 ioflupane scans were clustered using T-SNE, and one-way ANOVA was performed to compare HRV and SBR parameters. The analysis was repeated after the exclusion of patients taking angiotensin-converting enzyme inhibitors, given the known mechanism on autonomic function. Subsequent analysis showed a significant difference between the high-frequency domains of heart rate variability, asymmetry of the caudate SBR, and putamen-to-caudate SBR. Conclusion: Our results support the hypothesis that more imbalanced (specifically worse left-sided) neuronal loss results in greater AutD.

## 1. Introduction

Parkinsonian syndrome (PS) is a broad category of neurodegenerative movement disorders that includes Parkinson disease, multiple system atrophy (MSA), progressive supranuclear palsy, and corticobasal degeneration [1]. Differentiating between these disorders can be challenging and requires a combination of symptoms, physical exams, neurological exams, and imaging findings. Specifically, Parkinson disease (PD), which is the second most common neurodegenerative disorder, the most researched, and the most common PS, will, therefore, be preferentially discussed. PD is diagnosed according to the Movement Disorder Society Clinical Diagnostic Criteria for PD [2]. PD results from the loss of dopaminergic neurons in the substantia nigra, which project to the striatum [3]. The pathologic hallmark is intracellular aggregates of the toxic α-synuclein protein, which accumulates not only in neurons of the brainstem but also in the gut, olfactory bulb, and cortex [3,4,5]. PD can often be diagnosed clinically by the characteristic bradykinesia, unilateral rigidity, resting tremor, and postural instability [4].

A lesser-known but common non-motor symptom of PD is autonomic dysregulation (AutD) [6]. Symptoms associated with AutD can be debilitating and carry a host of potential non-motor dysfunctions, including cardiovascular and gastrointestinal problems [7]. The relationship between AutD and PD is not well understood; however, earlier development of AutD in PD patients led to more rapid progression of the disease and shorter survival [7]. However, it must be noted that the life expectancy of an individual with PD varies based on several factors, including the patient’s medical history.

Heart rate variability (HRV) measures the variations between heartbeats [8]. Various HRV parameters can be calculated, such as the standard deviation of the normal-to-normal beat intervals (SDNN), very low frequency (VLF), low frequency (LF), high frequency (HF), and total power (TP). A relatively higher HRV is found in normal, healthy people, while lower HRV is described in several disease processes [8]. HRV has also been implicated as a predictive measure of the severity of certain diseases, where a lower HRV is indicative of a worse prognosis. A decreased HRV was seen in diabetes mellitus [9] and cirrhosis, where a lower HRV was correlated with more severe disease and higher overall mortality [10]. Similarly, HRV has also been shown to be associated with a worse prognosis in PD. In a prospective cohort study, patients with a decreased HRV were associated with a higher risk of developing PD [11]. In another study, impaired HRV was shown to be strongly correlated with the duration of PD [6]. Studies have demonstrated the association of AutD with PD severity, progression, and survival [12] and that earlier AutD, demonstrated by reduced HRV, additionally predicts greater severity of PD and shorter survival.

Right and left sympathetic innervation of the heart have different effects on the QT interval. Blocking the right stellate ganglion increases the QT interval, while in contrast blocking the left stellate ganglion decreases the QT interval. Thus, it has been postulated that the left stellate ganglion may be a factor in arrhythmogenesis [13]. Indeed, a left stellate block has been used for ventricular arrhythmia treatment under certain clinical scenarios [14,15]. The asymmetry in cardiac innervation is important in the parasympathetic nervous system as well. The right vagus nerve predominantly regulates the sinoatrial node and the atria, while in contrast the left vagus nerve predominantly regulates the atrioventricular node and the ventricles [13]. 

In diagnostically challenging cases of possible PD, imaging with FDA-approved iodine-123 (I-123) ioflupane can be helpful. After injecting I-123 ioflupane, a gamma ray-emitting analog of cocaine with a high affinity for dopamine transporters in the striata, patients are imaged by single-photon emission computed tomography (SPECT) [16]. Normal I-123 ioflupane SPECT scans show radiotracer uptake in the striata, which are shaped like commas in the axial plane. Just as PD symptoms usually present initially on one side (hemi-Parkinsonism), the I-123 ioflupane SPECT imaging is also often asymmetric with greater loss of uptake contralateral to the side with worse symptoms [16]. Interestingly, studies have shown that the posterior putamen is the first to show decreased I-123 ioflupane uptake, and as the disease progresses, the decreased binding also progresses anteriorly [17].

Our objective was to explore the connection between neurodegeneration in the brain and AutD in PD. To accomplish this, quantitative parameters from dopamine transporter SPECT imaging were correlated with HRV parameters. Since PD causes asymmetric neuronal loss in the brainstem and cardiac autonomic innervation (also from the brainstem) is also asymmetric, we can divide the patient population into subgroups based on these differing patterns. We hypothesized that more severe left-sided neuronal loss would imbalance the autonomic innervation more severely, which would be reflected by worse HRV.

## 2. Materials and Methods

### 2.1. Patient Recruitment and Selection

Patients were recruited from the nuclear medicine clinic at the University of Arizona Medical Center on the day of their standard of care I-123 ioflupane SPECT scans, which were ordered to aid in the evaluation of suspected PS or PD. To minimize bias and simulate typical clinical practice, this pilot trial was an all-comers study in a real adult patient population referred for dopamine transporter imaging for movement disorders. Inclusion criteria were either a known or suspected diagnosis of PS or PD and an abnormal (by an expert radiologist read) I-123 ioflupane scan. The patients were presented with details of the study, risks and benefits were explained, and informed consent was obtained. The patients were then interviewed, and demographic information was collected. In addition, electronic medical records for each patient were reviewed to extract additional data such as current diagnosis of PS or PD (either definitively diagnosed or suspected), comorbidities, and current medications. Fifty-one patients initially consented; however, for various reasons, such as heart rate sensor malfunction, voluntary withdrawal from the study, and data corruption, six patients’ data could not be acquired. The exclusion criteria included medical comorbidities (such as a history of myocardial infarction, cardiac surgery or procedure, stroke, diabetes, hypertension, and cancer due to known confounding effects on HRV) and normal (by an expert radiologist read) I-123 ioflupane scans. None of the patients had a known diagnosis of dystonia or functional neurological disorders, which can often mimic PD [18,19] and would have been exclusion criteria if they were relevant. After applying the initial exclusion criteria, 13 patients’ data were analyzed. Subsequently, six patients taking angiotensin-converting enzyme (ACE) inhibitors were also excluded, and the final analysis was performed on seven patients.

These demographic data are outlined in Table 1. Table 2 presents patient-reported medications and the number of patients taking them.

### 2.2. HRV Data Acquisition

Once demographic data were collected, the patients were given a wearable biometric monitoring sensor, BioHarness3^TM^ (Figure 1, Zephyr Technology Corp, Annapolis, MD, USA). The BioHarness3 sensor was placed over the patient’s xiphoid process and adhered to the patient’s skin via electrocardiogram (ECG) pads (Figure 1). The sensor enables recording uni-channel ECG (250 Hz), respiration rate, accelerations, and approximate core body temperature [20,21]. Once the sensor adhered to the skin, the patients were instructed to lay in a supine position for 30 min, relax, and refrain from moving or speaking. ECG data were collected over the course of 30 min. The first and the last 5-min of the data stream were excluded for better signal quality. All ECG data were then processed to extract HRV features using Neurokit2, a Python script-based, user-friendly, open-source package [22], to ensure reproducibility. The obtained R-R intervals were then used to calculate the HRV time-domain measurements, including SDNN and RMSSD (root mean square of successive differences). SDNN is a global index of HRV and reflects longer-term circulation differences and circadian rhythms. Lower SDNN values indicate higher physiological stress responses, and higher SDNN values are linked to better wellbeing [21]. RMSSD is another method to quantify HRV, which reflects vagal tone. Higher values equate to higher parasympathetic activities or more relaxation [23]. Additionally, the R-R intervals were transformed by a Fourier transform to extract frequency-domain HRV features, such as HF power (0.15–0.4 Hz), LF power (0.04–0.15 Hz), and LF/HF ratio [17]. (Figure 2). The HF represents the parasympathetic regulations (relaxation indicator) of the heart. The LF/HF ratio indicates the balance between the sympathetic and parasympathetic activity of the heart. The higher LF/HF ratio, higher LF, and lower HF represent a higher balance toward sympathetic activation or a stressful condition [23]. We calculated both time and frequency domain parameters in every 5-min interval according to the guidelines of the European Society of Cardiology and the North American Society of Pacing and Electrophysiology [23].

### 2.3. I-123 Ioflupane SPECT Protocol and Quantification

Before injection, the patients were premedicated with Lugol solution to block the uptake of any free radioiodine by the thyroid gland. Three to four hours after the injection of approximately 185 MBq of I-123 ioflupane, projection data were obtained in a 128 × 128 matrix on a 2-head camera (ECAM SPECT from Siemens Medical Systems or Optima 640 SPECT/CT from General Electric Healthcare) mounted with low-energy high-resolution parallel-hole collimators. Projection data were acquired over 120 angles for approximately 30 min. The standard brain protocol was used, whereby the data were reconstructed using filtered back-projection without attenuation correction for the SPECT-only scanner and iterative reconstruction with CT attenuation correction for the SPECT/CT scanner.

Automated quantification was performed with the commercially available software DaTQUANT^TM^ (GE Healthcare; Chicago, IL, USA). The striatal binding ratios were calculated by determining the ratio of specific-striatal to nonspecific binding: [(mean counts in the striatal area—mean counts in the occipital cortex)/(mean counts in the occipital cortex)]. The software generates for each side the striatal binding ratios (SBR) for whole striatum, caudate, whole putamen, anterior putamen, and posterior putamen. Additional parameters included right and left asymmetry of the striatum, caudate, and putamen and the caudate to putamen ratio (Figure 3). I-123 ioflupane images were interpreted by a board-certified nuclear radiologist with 10 years of extensive experience interpreting I-123 ioflupane imaging.

### 2.4. Statistical Methods

Each subject’s data, a high dimensional dataset consisting of HRV parameters and I-123 ioflupane quantitative parameters, was unsuitable for visualization on a 2D map or for directly examining the correlation between specific HRV parameters and the I-123 ioflupane quantitative parameters. Therefore, we introduced T-distributed stochastic neighborhood embedding (t-SNE) [24], a dimension reduction and clustering visualization technique, to classify the subjects into different groups based on the subject data features.

t-SNE can convert a data set with hundreds of dimensions (high-dimensional) into a two/three-dimensional image by catching the local similarities in high-dimensional space while preserving the global structures as much as possible. Specifically, we first normalized all HRV and I-123 ioflupane scan parameters for the purpose of equal weights. Next, similarities between subjects in the high dimension were calculated based on the Gaussian distribution and the parameters’ Euclidean distances. The subjects were then randomly projected in a 2D space where similarities were measured with the Student t-distribution and the parameters’ Euclidean distances. Further, t-SNE continuously adjusted the distribution of subjects in 2D space to optimize the inconsistency between the pairwise similarities between corresponding subjects in the high dimension and those in the low dimension. Eventually, subjects with similar parameters are closer to each other and further from the separated clusters in the 2D map, revealing patterns in the data that were previously hidden. Different HRV variables and I-123 ioflupane scan parameters were selected until bright clusters formed in the t-SNE output. The t-SNE was implemented using Python v3.7 with the packages Scikit-learn v0.22 and SciPy v1.4. After the classification, the bootstrap method was utilized to estimate the mean of HRV measurements and I-123 ioflupane scans due to the limited sample size. The bootstrap method was performed in R, an open-source statistical programming language, and the seed was assigned as a constant to ensure reproducibility. The observed dataset was randomly resampled with a replacement 10,000 times. The resampled datasets are the same size as the observed dataset. Finally, the *p*-value was calculated based on the probability that the mean of the resampled datasets exceeds that of the observed dataset. Differences were considered significant at *p*-value < 0.05.

Therefore, the analyses were performed in two steps. First, the subjects were categorized into groups by the t-SNE algorithm. Second, bootstrapping was performed to test correlations between caudate symmetry and HRV measurements across the classified groups.

## 3. Results

### 3.1. Patients

A total of 51 patients provided their consent to participate in the study and were enrolled. However, over the course of the study, multiple patients were excluded from the analysis due to medical comorbidities such as a history of myocardial infarction, cardiac surgery or procedure, stroke, diabetes, hypertension, and cancer. These demographic data are outlined in Table 1. Table 2 presents the patient-reported medications and the number of patients taking them. After exclusion criteria, 13 patients’ data with abnormal (by an expert radiologist read) I-123 ioflupane scans were utilized in the final analysis.

### 3.2. Heart Rate Variability

The mean value of HRV was obtained (Figure 4), the time and frequency domain HRV features were extracted, and the quantitative data obtained from the I-123 ioflupane scan were compared using various statistical methods.

One of the measures tested was the lower value of the posterior putamen since the posterior putamen is the earliest affected in PD compared to the different HRV metrics. The data showed a linear fit with a 95% confidence interval with all the metrics. Of these, caudate asymmetry and the HF power metric had the highest R^2^ value and were significant, while posterior putamen and HF power did not show a significant correlation (Figure 5).

The data from patients with abnormal I-123 ioflupane were clustered using T-SNE. The HRV parameters were compared to all the available striatal binding quantitative data via one-way ANOVA. Most of the quantitative SBR values were not found to be significantly different across the clustered groups. However, the HRV parameters were found to have significant differences between the groups. Before comparison, patients with a history of myocardial infarction, cardiac procedures, diabetes, stroke, hypertension, and cancer were excluded from the analysis. Subsequently, the data was reclassified by T-SNE and reanalyzed via one-way ANOVA with similar results (Figure 6: Group A, *n* = 6; Group B, *n* = 4; and Group C, *n* = 3). Given the often-severe loss of uptake in the posterior putamen even in cases of early PD, the analysis was repeated using the anterior putamen, caudate, and, finally, the entire striatum. However, no significant difference was found in any of these measures.

Subsequently, in order to minimize the confounding effects of medications on autonomic function, the patients taking angiotensin-converting enzyme (ACE) inhibitors were excluded. These patients were clustered using T-SNE, and the patients taking ACE inhibitors were excluded. The remaining patients were then analyzed (Group A, *n* = 5; Group B, *n* = 2; Group C, all three excluded). The high-frequency domain of the heart rate variability, right versus left asymmetry of the caudate uptake on the I-123 ioflupane, and putamen-to-caudate striatal binding ratio on the right showed a significant correlation with the HF-power *p*-value of 0.0456 (Figure 7a), caudate asymmetry *p*-value of 0.0165 (Figure 7b), and right putamen/caudate ratio *p*-value of 0.0341 (Figure 7c). The remainder of the computed metrics did not show a significant difference between the two groups.

The raw data showed most patients’ left caudate SBR to be lower than the right, and only two patients’ right caudate SBR was lower than the left.

## 4. Discussion

The autonomic dysfunction in PD has been found to have a laterality bias and hemispheric dominance [16,17,25]. Similarly, asymmetric cardiac autonomic innervation has also been shown and utilized clinically [13,14,15]. In our study, the HRV parameters were fitted to the SBR of the posterior putamen, the first region of the striatum to be affected in PD. All HRV metrics had a positive correlation to posterior putamen (PP), but the highest correlation was with HF-power. Cluster analysis showed three distinct clusters with HF-power, RSMMD, and PP. Initial ANOVA did not show a statistically significant difference in PP across all three groups (A, B, and C) but was statistically different between groups A and B. This led us to examine the subjects in group C, and it was uncovered that all three of the subjects in group C were taking an ACE inhibitor (some subjects from groups A and B were also taking ACE inhibitors and were also excluded). After this exclusion, only subjects in groups A and B remained for the analysis. Subsequent ANOVA showed a statistically significant difference between all groups. The initial lack of significance was likely due to the confounding mechanism of action of ACE inhibitors on autonomic function and, thus, logically could be used as an exclusion criterion in future studies. The significant difference between the right putamen/caudate ratio is likely explained by the advanced disease of these patients, where the disease had progressed beyond the posterior putamen and had involved the anterior putamen. The caudate to putamen ratio was significant only on the right side. Most patients’ left caudate SBR value was lower than the right, and the putaminal asymmetry was lower than the caudate asymmetry. We believe this was because the patients’ disease was advanced enough to markedly affect the putamen bilaterally, but since the caudate is affected last, it represents the most robust marker of asymmetry. The more severely affected left side supports our hypothesis that more severe neuronal loss on the left would have a more profound effect on the HRV.

Many lessons were learned from this trial for future research. We demonstrated the feasibility of recruiting patients presenting for their standard of care I-123 ioflupane imaging. This recruitment method has both advantages and disadvantages. A major advantage is the convenience for the patient since the HRV measurements are accomplished during the one-hour period of waiting after drinking the iodine solution and before the radiopharmaceutical injection, but this limits the amount of ECG time. The HRV measurements and the I-123 ioflupane imaging are also performed near-simultaneously, ensuring that the HRV measurements are coupled temporally to the imaging. This recruitment strategy likely has less bias than recruiting from neurology clinics since little is known about the patients being recruited, but the yield of confirmed PD patients is much lower. Also, it is difficult to pre-screen for confounding factors like stroke, arrhythmias, heart failure, and the use of ACE inhibitors. Our recruitment method also required a nuclear medicine clinic with a busy I-123 ioflupane imaging program and, thus, a strong referral base.

Future work could include ECG measurements for longer periods, perhaps even continuous 24-h monitoring, which could provide more accurate HRV analyses. HRV could also be combined with other measures of AutD to provide a more complete assessment of the sympathetic/parasympathetic balance. Artificial intelligence methods could also be used to further study the connection between neurodegeneration in PD and AutD. Convolutional neural networks could be used to analyze images without restriction to pre-specified regions in quantification software. Another potential avenue for future work could include the evaluation of additional clinical data (such as the time from symptom onset) and their effect on disease severity.

An important limitation of this trial was the small sample size, especially after exclusion criteria were applied. Another limitation was patients’ medications for their preexisting comorbidities, such as ACE inhibitors. The importance of excluding patients who were taking ACE inhibitors was shown during our analysis. Other medications may also influence the autonomic function and could thus also be confounding factors. However, an important caveat of medications is that patients cannot practically be asked to stop the use of these medications without putting patients at risk. Another challenge faced was the presence of comorbidities. Several confounding medical problems that are known to affect HRV and require exclusion, such as stroke, prior cardiac surgery, cardiac arrhythmias, and heart failure, are frequently encountered in the age group where PD is most prevalent. In addition to cerebrovascular and cardiovascular comorbidities, neuropsychiatric comorbidities such as dementia or mood disorders could also be confounding factors and should be stratified in future studies with larger patient populations. Another challenge faced during HRV measurement was that there was no way a priori to identify which patients were going to be abnormal before the I-123 ioflupane scan.

Despite these limitations and challenges, the trial highlighted several possible avenues for further research. Since I-123 ioflupane SPECT could potentially stratify the severity of AutD in PD patients, imaging could be incorporated into the clinical management of PD patients beyond the initial diagnostic work-up. The pathophysiology of PD is heterogeneous with the involvement of nigrostriatal and other neural pathways [26]. There may be input from the right hemisphere-limbic system into the autonomic system, which could be affecting the autonomic regulation of the heart. Similarly, complex input from cortical and subcortical structures involved in emotional responses may also influence autonomic balance and can be researched in the future.

## Figures and Tables

**Figure 1 sensors-22-05055-f001:**
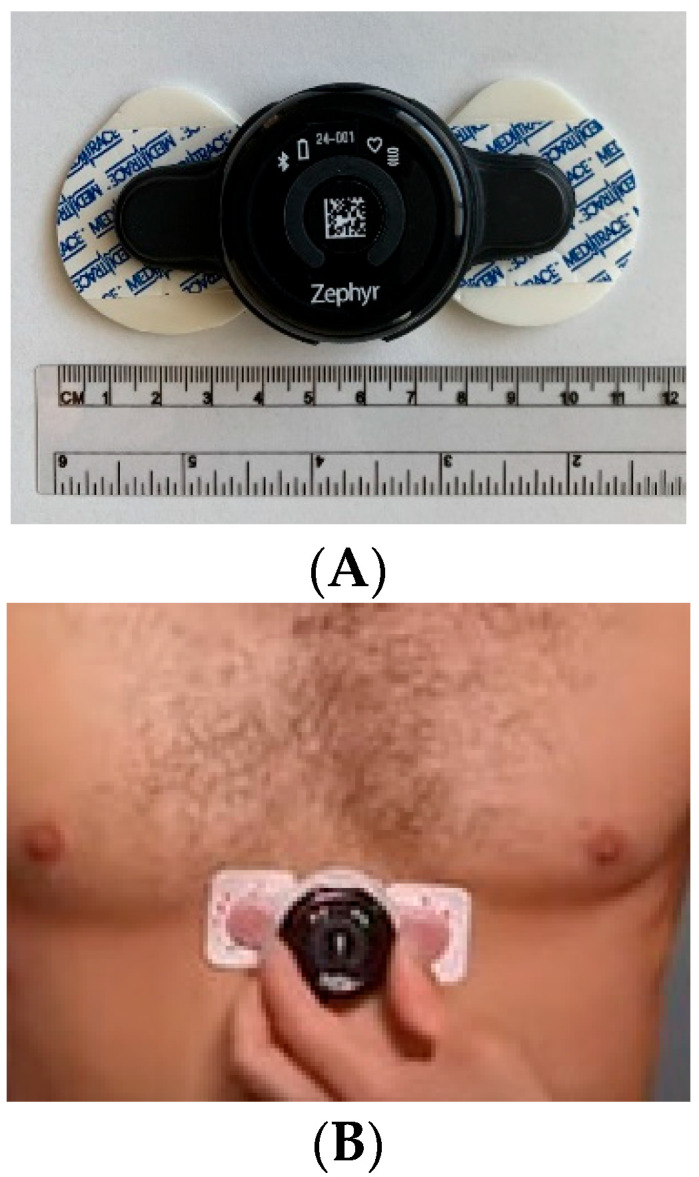
Image of the sensor and ECG pads (**A**) adhered to the patient’s chest over the xiphoid process (**B**).

**Figure 2 sensors-22-05055-f002:**
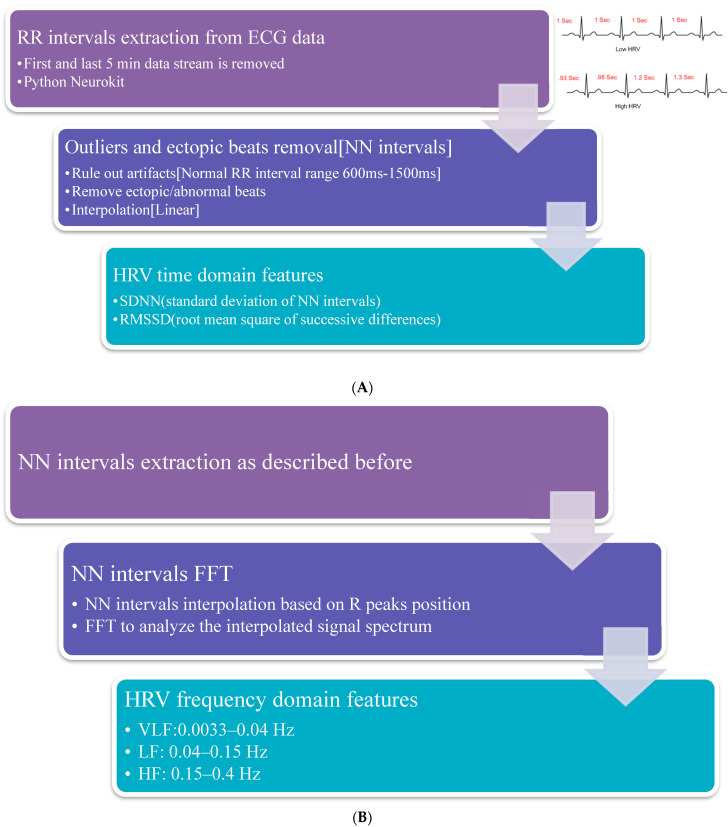
Extraction of HRV metrics from the ECG data collected from the biosensor. (**A**) Time domain metrics. (**B**) Frequency domain metrics.

**Figure 3 sensors-22-05055-f003:**
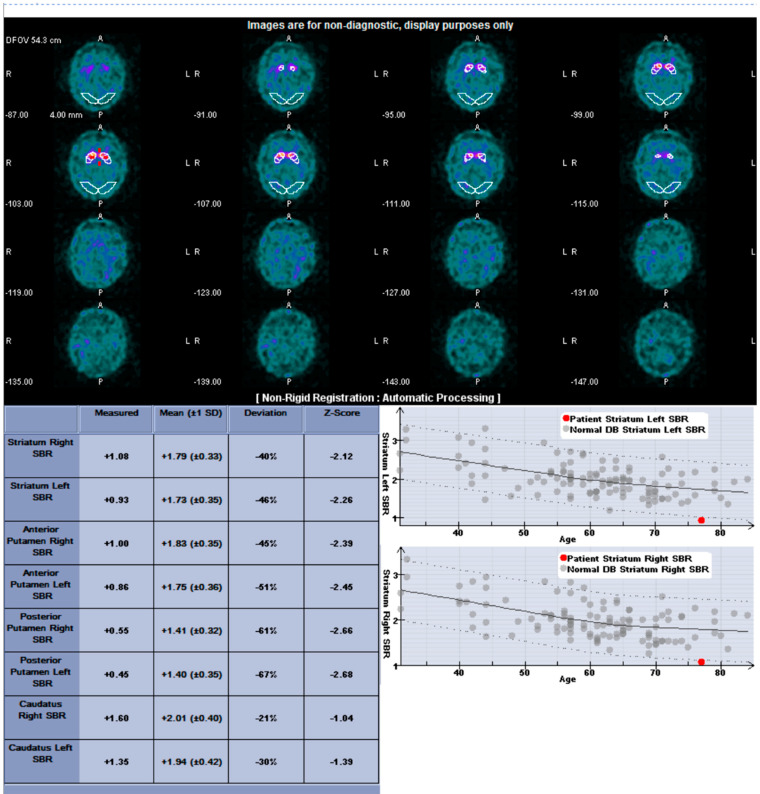
DaTQUANT output for patient 16, as an example. (**Top**): SPECT images with region of interest (ROI) over the striata and background ROIs over occipital region. (**Bottom left**): Table of quantitative parameters. (**Bottom right**): Graph of quantitative measures compared to a population of normal subjects. The red dot denotes the patient’s striatal value, which is abnormally decreased (solid line represents the average for normal subjects and dashed lines represent 2 standard deviations for normal subjects).

**Figure 4 sensors-22-05055-f004:**
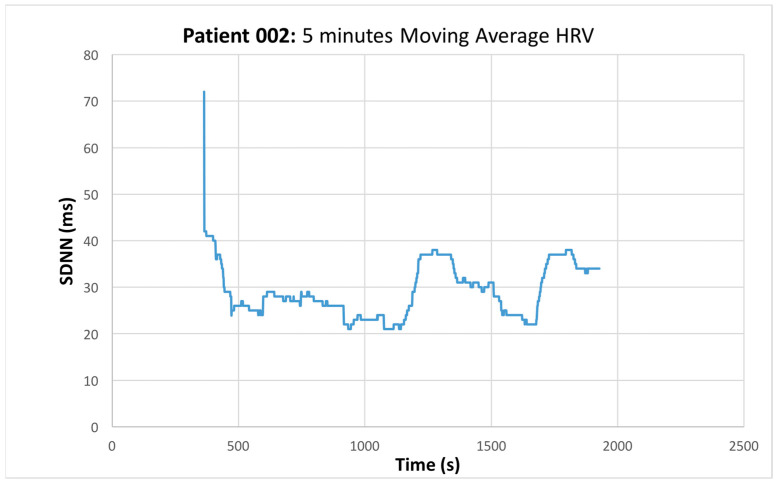
SDNN (ms) versus Time (s). Representative graph of the moving average measured HRV.

**Figure 5 sensors-22-05055-f005:**
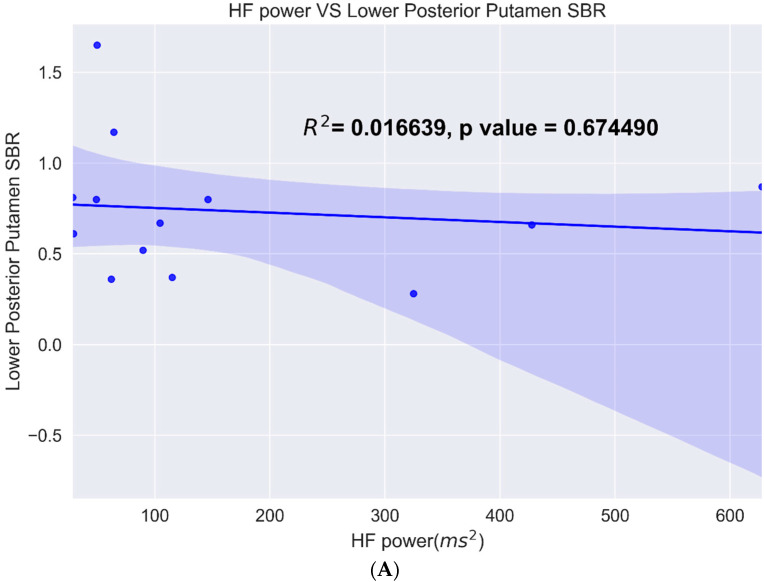
(**A**) Linear fit of posterior putamen binding ratio versus various HF power. (**B**) Linear fit of caudate asymmetry binding ratio versus HF power.

**Figure 6 sensors-22-05055-f006:**
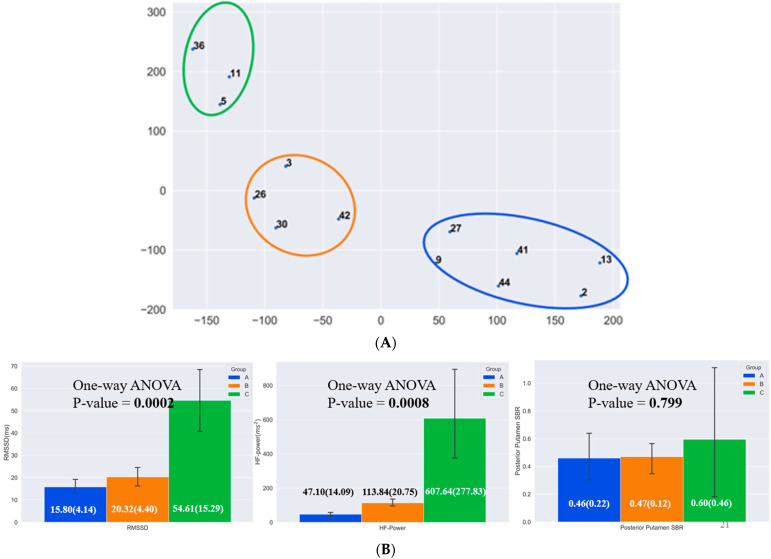
(**A**) T-SNE clustering of patients by posterior putamen SBR (PP), HF-power, and RMSSD. Blue oval: Group A; Orange oval: Group B; Green oval: Group C. (**B**) One-way ANOVA results of PP, HF-power, RMSSD show significant difference between the groups for HRV parameters only and not with PP.

**Figure 7 sensors-22-05055-f007:**
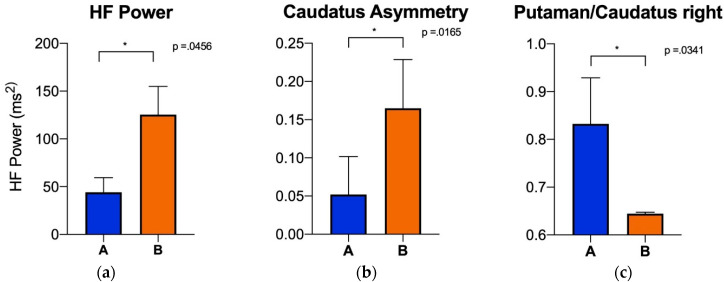
Groups were clustered first using T-SNE. Subsequently, patients taking ACE inhibitors were excluded from the analysis to minimize confounding effects on autonomic function (which included all the patients in Group C, and thus, only Groups A [blue] and B [orange] are shown). The analysis shows significant difference between the two groups for all parameters. (**a**): HF (high frequency; *p*-value = 0.0456) power; (**b**): caudate asymmetry (*p*-value = 0.0165); (**c**): right putamen/caudate ratio (*p*-value = 0.0341). Error bars indicate the standard deviation, * indicates *p*-value < 0.05.

**Table 1 sensors-22-05055-t001:** Demographics and comorbidities of the entire patient population. The patients that were analyzed have been identified by colored boxes, and those of the analyzed patients taking angiotensin-converting enzyme (ACE) inhibitors are marked by asterisks. Blue box—Group A; Orange box—Group B; Green box—Group C. Six patients’ data could not be acquired and are, thus, not included in this table (patients 4, 6, 10, 14, 15, and 28). PD—Parkinson disease; HTN—hypertension; PS—parkinsonian syndrome; COPD—chronic obstructive pulmonary disease; MSA—multiple system atrophy; GERD—gastroesophageal reflux disease; ADHD—attention deficit/hyperactivity disorder; BPH—benign prostatic hyperplasia; CABG—coronary artery bypass graft; MI—myocardial infarction; CKD—chronic kidney disease.

Patients						
	Sex (M/F)	Age (years)	Weight (kg)	Height (m)	BMI (kg/m^2^)	Co-Morbidities
**1**	F	75	116.12	1.74	38.36	PD, Headaches, Cancer, COPD, HTN
**2**	M	84	77.11	1.83	23.06	Diabetes, Cancer
**3**	F	68	81.65	1.7	28.19	Asthma, Vertigo,
*** 5**	M	74	77.56	1.73	26	PD, Neuropathy, HTN, Pain
**7**	M	73	95.25	1.8	29.29	MSA, GERD, Hypercholesterolemia
**8**	M	53	81.65	1.83	24.41	ADHD, Anxiety, Seizures
*** 9**	M	70	69.4	1.78	21.95	Hypercholesterolemia, Diabetes, HTN, Dementia
*** 11**	M	71	86.18	1.78	27.26	PD, GERD, HTN, COPD, BPH
**12**	F	62	63.5	1.6	24.8	Pain, ADHD, Anxiety, Mental Health, Depression, GERD, Cancer, Pancreatitis
**13**	M	62	71.67	1.75	23.33	PD
**16**	M	78	83.91	1.78	26.54	PS, HTN, Mood Disorder, Dementia, Sleep Apnea, Neuropathy
**17**	F	76	63.5	1.63	24.03	PS, Cancer, HTN, Depression, Colitis, Raynaud’s Syndrome, Hypercholesterolemia, Pain
**18**	F	81	53.52	1.55	22.3	Depression, Lupus, Stroke, HTN, Hypercholesterolemia
**19**	M	69	104.33	1.83	31.19	
**20**	M	80	73.48	1.68	26.15	Hypercholesterolemia, Diabetes, HTN
**21**	F	53	108.86	1.65	39.94	Angina, HTN
**22**	F	75	61.23	1.68	21.79	GERD, Pain, Fibromyalgia, Hypercholesterolemia
**23**	M	75	86.18	1.65	31.62	
**24**	M	82	99.79	1.75	32.49	HTN, Osteoarthritis, BPH, GERD, Gout
**25**	M	73	83.01	1.8	25.52	GERD, Heart Disease, Hypercholesterolemia, Asthma
*** 26**	F	84	64.41	1.6	25.15	Pain, Arthritis, HTN
**27**	F	73	56.7	1.5	25.2	PS, Supranuclear Palsy, Depression, Arthritis
**29**	M	65	72.57	1.68	25.8	HTN, Stroke, Neuropathy, Lymphoma
**30**	M	83	60.33	1.73	20.2	HTN, Hyperlipidemia, Seizures
**31**	M	74	77.11	1.63	29.2	Depression, Tremors, HTN
**32**	F	71	81.65	1.57	32.9	Hypothyrodism, Migraines, Sleep Apnea, GERD, Depression, Hypercholesterolemia
**33**	F	71	58.51	1.68	20.8	PS, Cancer, Hypercholesterolemia, HTN, Dementia, Depression, Anxiety, Hypothyroidism
**34**	M	73	86.18	1.75	28.1	HTN, Depression, GERD
**35**	M	72	73.48	1.68	26.1	HTN, GERD, BPH, Depression, Hypercholesterolemia
*** 36**	M	41	102.97	1.85	29.9	PS, HTN
**37**	F	71	52.16	1.65	19.1	Dementia, Tremors, Hypercholesterolemia, Neuropathy, Diarrhea
**38**	M	73	127.01	1.88	35.9	HTN, BPH, Hypercholesterolemia, Sleep Apnea, CABG
**39**	M	77	84.82	1.75	27.6	Tremor, Achalasia, Cancer, CABG, Hypothyroidism, Hypercholesterolemia, Osteoporosis
**40**	F	66	95.25	1.52	41	MI, Diabetes, Osteoarthritis, CKD, Hypothyroidism, Depression, Hypercholesterolemia, GERD, BPH
**41**	M	64	108.86	1.78	34.4	PS, HTN, Hypercholesterolemia, Hypothyroidism
*** 42**	M	76	85.28	1.7	29.4	HTN, Neuropathy, Pain, Gait Instability
**43**	M	69	81.65	1.73	27.4	PD, Stroke, HTN, Hypercholesterolemia, BPH, Depression, Arthritis, Diabetes
**44**	M	84	81.65	1.78	25.8	BPH, Pain, Dementia, Neuropathy, Glaucoma
**45**	M	75	87.54	1.85	25.5	Diabetes, Glaucoma, GERD
**46**	M	77	62.5			HTN
**47**	F	60	67.59	1.63	25.6	HTN, Dystonia, Arthritis
**48**	M	78	76.2	1.7	26.3	HTN, Stroke, Abdominal Aneurysm
**49**	M	68	72.57472	1.73	24.3	
**50**	M	74	123.83062	1.75	40.3	
**51**	M	75	89.36	1.73	29.7	HTN, Cancer, Hypothyroidism

**Table 2 sensors-22-05055-t002:** Patient-reported medications with number of total patients and analyzed patients taking each medication.

Medications		Number of Patients	Number of Analyzed Patients
**Antihypertensives**	**β-blocker**	14	1
	**ACE-I/ARB**	19	6
	**Other**	12	2
**PD/Parkinsonian/Dementia**	**Levadopa/Carbidopa**	12	4
	**Cholinergic**	4	2
	**Other**	6	1
**Additional Medications**	**Gabapentin/Anti-seizure**	12	4
	**Opiate/Opioid**	5	1
	**Psychiatric Medications**	20	2
	**β-agonist**	2	2
	**Steroid**	3	
	**NSAIDs**	17	5

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
