# Peer review of "Pilot Study for Correlation of Heart Rate Variability and Dopamine Transporter Brain Imaging in Patients with Parkinsonian Syndrome"

_sensors, 2022, doi:10.3390/s22135055_

Round 1

Reviewer 1 Report

Warhadpande et al. investigated the correlation between heart rate variability and DAT Binding. They report a negative correlation between left caudate tracer uptake and HRV. Their results indicate that an asymmetrical DAT uptake is responsible for HRV.

Major points:

-         In the title it is noted that brain imaging, indicate a parkinsonian syndrome. The whole text the authors implicate that the results observed came from patients suffereing from Parkinson’s disease. As long as no diagnostic criteria were applied, it is open, what disease identities were examined.

-         Even when the intention is to report a pilot study, the number of patients are insufficient to draw a preliminary conclusion. 13 patients were left in the analysis, unfortunately 6 of them were taking angiotensin converting enzyme inhibitors, that interfere with HRV. Two patients were on ß-agonist. Statistical setting (e.g. ANOVA) do not work sufficiently in such a low sample size.

-         If the study was designed as a prospective analysis, an ethical vote has to be included, please register the study to make clear the primary endpoints at the beginning.

Reviewer 2 Report

The authors reported an interesting preliminary study about the correlation of Heart Rate Variability and Dopamine Transporter Brain Imaging in Patients with Parkinsonian Syndrome. I have some comments to the authors:

- In the introduction please better describe the part regarding the diagnosis of PD, with particular references to the MDS diagnostic criteria that you should mention: 

Postuma RB, et al. MDS clinical diagnostic criteria for Parkinson's disease. Mov Disord. 2015 

- In the methods is not clear how you have selected the patients from the pool of 51. It would be easier if the authors specify the inclusion and exclusion criteria for the final analysis. Please also include among the exclusion criteria dystonia (especially tremor type) and functional neurological disorders (that can often mimic PD). Here some references that the authors should add in the manuscript: 

Defazio G, et al. Idiopathic Non-task-Specific Upper Limb Dystonia, a Neglected Form of Dystonia. Mov Disord. 2020 

Hallett M, Aybek S, Dworetzky BA, McWhirter L, Staab JP, Stone J. Functional neurological disorder: new subtypes and shared mechanisms. Lancet Neurol. 2022 Jun;21(6):537-550. 

- Please specify all the abbreviation in table 1

- In Table 2 you should describe psychiatric medication in details. The authors need to specify if any of the patients was on antipsychotic meds (which are an important cause of iatrogenic extrapyramidal syndrome). Moreover why some patients were on Levodopa/Carbidopa? Can you provide the LEDD for those patients?

- Within this context it would be better to specify also other clinical data of the patients, such as the time elapsing from the symptoms onset and the study enrollment.

- Another important point is the definition of parkinsonism in you group, indeed it is not clear whether the patients received a diagnosis of PD or not. Please add more info about this point, at least for the 13 patients.

Reviewer 3 Report

Thanks for recommending me as a reviewer. In this paper, since PD causes asymmetric neuronal loss in the brainstem and cardiac autonomic innervation (also from the brainstem) is also asymmetric, authors can divide the patient population into subgroups based on these differing patterns. In this study, authors hypothesized that more severe left-sided neuronal loss would imbalance the autonomic innervation more severely which would be reflected by worse HRV. If authors complete minor revisions, the quality of the study will be further improved.

1. The introduction section is well written. However, the purpose of the study is unclear. If the author clearly describes the purpose of the research, it can help readers understand it.

2. line 115-123: In the Methods section, the authors should more specifically characterize the Parkinson's patients recruited for this study.

3. The authors should be more specific about the t-SNE algorithm in the Methods section.

Round 2

Reviewer 2 Report

The authors have addressed all the points.